# The ceramide synthase 2b gene mediates genomic sensing and regulation of sphingosine levels during zebrafish embryogenesis

Karen Mendelson[1,2], Suveg Pandey[1], Yu Hisano[3,4], Frank Carellini[1], Bhaskar C Das[5], Timothy Hla[3,4]*, Todd Evans[1]*

[1]Department of Surgery, Weill Cornell Medical College, Cornell University, New York, United States; [2]Center for Vascular Biology, Department of Pathology and Laboratory Medicine, Weill Cornell Medical College, Cornell University, New York, United States; [3]Vascular Biology Program, Department of Surgery, Boston Children's Hospital, Boston, United States; [4]Harvard Medical School, Boston, United States; [5]Department of Medicine, Division of Nephrology, Icahn School of Medicine at Mount Sinai, New York, United States

*For correspondence:
timothy.hla@childrens.harvard.edu (TH);
tre2003@med.cornell.edu (TE)

Competing interests: The authors declare that no competing interests exist.

**Abstract** Sphingosine-1-phosphate (S1P) is generated through phosphorylation of sphingosine by sphingosine kinases (Sphk1 and Sphk2). We show that *sphk2* maternal-zygotic mutant zebrafish embryos (*sphk2[MZ]*) display early developmental phenotypes, including a delay in epiboly, depleted S1P levels, elevated levels of sphingosine, and resistance to sphingosine toxicity. The *sphk2[MZ]* embryos also have strikingly increased levels of maternal transcripts encoding ceramide synthase 2b (Cers2b), and loss of Cers2b in *sphk2[MZ]* embryos phenocopies sphingosine toxicity. An upstream region of the *cers2b* promoter supports enhanced expression of a reporter gene in *sphk2[MZ]* embryos compared to wildtype embryos. Furthermore, ectopic expression of Cers2b protein itself reduces activity of the promoter, and this repression is relieved by exogenous sphingosine. Therefore, the *sphk2[MZ]* genome recognizes the lack of sphingosine kinase activity and up-regulates *cers2b* as a salvage pathway for sphingosine turnover. Cers2b can also function as a sphingolipid-responsive factor to mediate at least part of a feedback regulatory mechanism.
DOI: https://doi.org/10.7554/eLife.21992.001

## Introduction

Sphingosine-1-phosphate (S1P) is a lysophospholipid mediator generated through sphingomyelin metabolism, best known as a receptor-dependent regulator of lymphocyte trafficking and vascular homeostasis. However, S1P signaling also plays key roles during embryogenesis, including developmental angiogenesis, cardiogenesis, limb development and neurogenesis (*Proia and Hla, 2015*). S1P binds to five G-protein-coupled receptors (S1P$_{1-5}$) that are widely-expressed and function either alone or together to regulate various developmental and physiological functions (*Mendelson et al., 2014*). S1P can only be generated via phosphorylation of sphingosine by sphingosine kinase, encoded by two genes (*sphk1* and *sphk2*) that are highly conserved in vertebrates. In zebrafish, *sphk2* is expressed during the earliest stages of embryogenesis, including as a maternal transcript (*Mendelson et al., 2015*). Mouse embryos tolerate knockout of either *Sphk1* or *Sphk2* (*Allende et al., 2004*), which implies that *Sphk1* and *Sphk2* function redundantly and can compensate for each other's loss. Indeed, global or hematopoietic-specific (erythrocyte and platelet) knockout of both *Sphk1* and *Sphk2* is embryonic lethal between E11.5–13.5 due to vascular

**eLife digest** Humans and other animals need a variety of fat molecules known as lipids to survive and grow. For example, members of a large family of lipids called sphingolipids play crucial roles in many cell processes and are essential for tissues and organs to form as animal embryos develop. A molecule called sphingosine is the basic building block of most sphingolipids. An enzyme known as sphingosine kinase 2 (Sphk2) converts sphingosine into a sphingolipid called sphingosine 1-phosphate, which is released from cells and regulates communication between cells. Another enzyme called ceramide synthase 2b (Cers2b) converts sphingosine into ceramide, which is required for the membrane surrounding individual cells to work properly.

Excess levels of particular sphingolipids can be toxic to cells so animals carefully control the levels of these molecules in their bodies. However, it was not clear how animal cells sense the molecules and regulate their production and breakdown. Mendelson et al. addressed this question by studying mutant zebrafish embryos that fail to produce Sphk2 and thus produce little or no sphingosine 1-phosphate.

It was thought that the mutant zebrafish may experience the toxic effects of high levels of sphingosine because they are unable to convert it to sphingosine 1-phosphate. However, the experiments show that the mutant embryos are able to avoid this toxicity by increasing the production of the Cers2b enzyme. Somehow the gene that encodes this enzyme is able to sense high levels of sphingosine and responds by increasing its own production, which in turn increases the conversion of sphingosine into the non-toxic ceramide. Further experiments found that this mechanism for sensing and removing excess sphingosine is already operating at an earlier stage of development when the zebrafish is just an egg.

Further work is needed to understand exactly how this sphingosine sensing mechanism works. These findings may help us to develop new treatments for diseases caused by an imbalance in sphingolipids, such as diabetes and cancer. This work may also help to increase the success of in vitro fertilization (IVF) and other fertility therapies by enhancing the survival of human egg cells.
DOI: https://doi.org/10.7554/eLife.21992.002

developmental defects (*Gazit et al., 2016*; *Mizugishi et al., 2005*; *Xiong et al., 2014*) that are similar to those observed in mice lacking all three receptors $S1P_{1-3}$ (*Liu et al., 2000*). Because of this early lethality, functions for sphingosine kinases in murine embryogenesis are otherwise not well characterized.

Zebrafish genetics has contributed much of our understanding for how S1P signaling regulates collective cell migration during the process of primordial heart tube formation from two anterior lateral mesoderm progenitor fields. A point mutation in the gene *miles apart* (*mil*), which encodes $S1P_2$, prevents cardiac precursor cells from migrating to the midline, resulting in formation of two separate defective hearts or *cardia bifida* (*Kupperman et al., 2000*). The mutants *two of hearts* (*toh*) and *ko157* phenocopy the *mil* cardiac defect and are caused by recessive mutations in the sphingolipid transporter gene, *spinster2* (*spns2*) (*Kawahara et al., 2009*; *Osborne et al., 2008*). A model consistent with the genetics is that *spns2* functions in the yolk syncytial layer (YSL, analogous to mammalian extra-embryonic endoderm) to transport S1P to embryonic endoderm where it activates $S1P_2$ signaling, coupled to $G\alpha_{13}$ that subsequently activates RhoGEF to regulate endodermal convergence (*Ye and Lin, 2013*). In this way, the $S1P_2$/$G\alpha_{13}$/Rho pathway activates cell-surface integrins and fibronectin matrix assembly (*Zhang et al., 1999*) and provides endoderm with the matricellular cues recognized by myocardial precursor cells to migrate collectively towards the midline. Receptor signaling is mediated in part through YAP1-dependent expression of CTGF in S1P-activated endoderm (*Fukui et al., 2014*).

We (*Mendelson et al., 2015*) and others (*Hisano et al., 2015a*) generated maternal zygotic mutants for *sphk2* (*sphk2^{MZ}*) and showed that these embryos phenocopy null zygotic mutants of *s1pr2* and *spns2*. The *sphk2* transcripts can be either maternally or zygotically provided, but without either, essential migrational cues fail to be transmitted to precardiac mesoderm and formation of the heart tube fails. Sphk2 activity in the YSL, presumably acting in concert with the Spns2 transporter, establishes a gradient of extracellular S1P essential for the provision of endodermal cues that

are recognized by overlapping sheets of migrating precardiac mesoderm. Endoderm convergence/heart tube formation is the earliest known developmental function for receptor-dependent S1P signaling. However, the studies do not rule out other receptor-independent functions for Sphk2.

We have since re-evaluated the *sphk2$^{MZ}$* embryos and discovered phenotypes that are distinct from those found by disruption of *spns2* or *s1pr2*. These phenotypes appear to be caused not by loss of S1P, but rather by the excessive accumulation of the precursor substrate sphingosine in the absence of kinase. This led us to the discovery that embryos lacking sphingosine kinase activity tolerate the potentially toxic excess sphingosine because they upregulate a specific ceramide synthase (Cers) gene, as a salvage pathway to normalize sphingosine levels. We mapped a DNA region that mediates the transcriptional response on the *cers2b* promoter. It has long been noted that Cers enzymes, in addition to encoding a Lag motif important for enzymatic activity that metabolizes sphingosine to ceramide, also encode a homeobox domain, typically used for DNA-binding by the HOX class of transcription factors (*Levy and Futerman, 2010*). Indeed, we show that Cers2b, probably by an indirect mechanism, can modulate the expression of its own promoter, and this activity requires the HOX domain and is sensitive to sphingosine levels. Our study identifies cis-linked sphingolipid transcriptional response sequences, and reveals a sensing mechanism controlling sphingolipid levels by a metabolic enzyme that participates in a feedback program.

## Results

### Maternal-zygotic mutant embryos have previously unrecognized early developmental phenotypes

Upon close examination of developing embryos, we noticed that *sphk2$^{MZ}$* embryos display a subtle but consistent developmental delay during gastrulation, compared to time-matched wildtype embryos, particularly obvious around 7–8 hr post fertilization (hpf) (*Figure 1A–D*). The developmental delay observed in *sphk2$^{MZ}$* embryos was recapitulated in wildtype embryos treated with 3 μM sphingosine (*Figure 1E,F*). In contrast, wildtype embryos were unaffected by culturing in the presence of ceramide (20 μM), S1P (20 μM), or N-acetyl-D-sphingosine (20 μM) (*Figure 1G–I*). Quantitative lipidomics profiling confirmed that the mutant embryos are depleted of S1P to essentially undetectable levels. However, this analysis also revealed that the mutant embryos have excessive levels of sphingosine and total long chain ceramides (*Figure 1J,K*), likely due to substrate backup caused by lack of sphingosine kinase and reverse reactions that form ceramide. These observations indicate that in the absence of Sphk2, embryos accumulate excessive levels of the precursor substrate sphingosine, associated with a developmental delay during gastrulation. This is however ultimately tolerated by embryos, and they recover, until displaying later phenotypes in endoderm convergence and heart formation.

Although *sphk2$^{MZ}$* mutant embryos accumulate sphingosine and display developmental delay, we discovered that they are relatively resistant to exogenous sphingosine-induced toxicity. Treatment of wildtype embryos with 5 μM D-*erythro*-sphingosine (the active naturally occurring isomer) or 5 μM L-*erythro*-sphingosine (an inactive isomer) caused developmental collapse during epiboly, a stage of collective cell migration of epiblast cells during gastrulation. At around 50–60% epiboly, in 100% of the treated embryos, epiboly arrests, the epiblast pinches off the yolk cell, and the yolk membrane ruptures, leading to embryonic lethality (*Figure 2A–C*). Treatment with sphingosine analogues (psychosine, dimethylsphingosine, and safingol) recapitulated this epiboly phenotype (*Figure 2D–F*). Treatment with two chemical inhibitors of sphingosine kinases, SKI (10 μM) or BT-190 (3 μM) also recapitulated the same embryonic lethal phenotype (*Figure 2G,H*). In contrast, *sphk2$^{MZ}$* embryos given the same treatments do not fail epiboly and (other than mild delay) develop normally (*Figure 2I*) until they display later endoderm/cardiac defects. Therefore, accumulation of sphingosine above a certain threshold is normally incompatible with embryonic development, yet tolerated in *sphk2$^{MZ}$* mutant embryos.

### Epiboly defects caused by excess sphingosine are associated with a disturbed cytoskeleton

The developing zebrafish blastula is comprised of three cell layers: an extra-embryonic transient epithelial monolayer known as the enveloping layer (EVL), an extra-embryonic yolk syncytial layer (YSL),

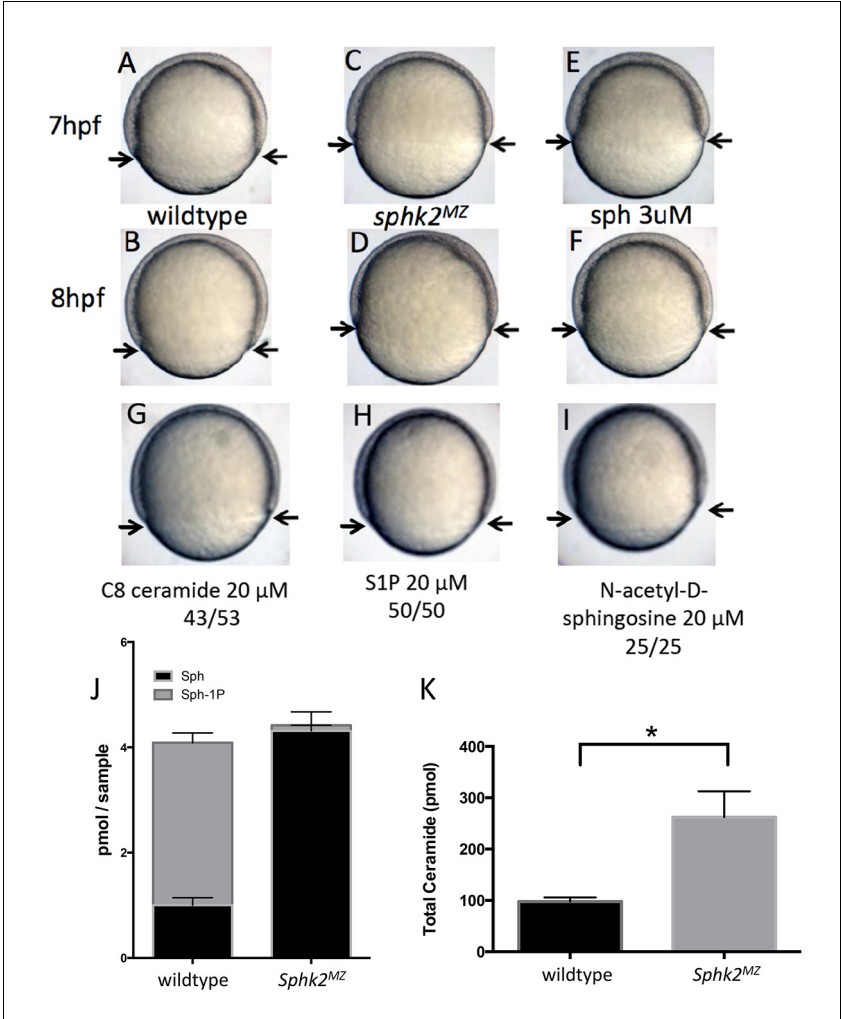

**Figure 1.** The *sphk2^MZ* embryos show developmental delay associated with enhanced levels of sphingosine. (**A,B**) Compared to time-matched representative wildtype embryos observed at 7 or 8 hr post fertilization (hpf), the *sphk2^MZ* embryos (**C,D**) display a developmental delay during epiboly, denoted by arrows indicating the leading edge of the blastoderm. (**E, F**) Wildtype embryos treated at 2 hpf with 3 µM sphingosine (sph) were similarly developmentally delayed. Treatment of wildtype embryos with G) C8 ceramide (20 µM), (**H**) S1P (20 µM), or (**I**) N-acetyl-D-sphingosine (20 µM) had no adverse effects on epiboly. A-I show representative embryos, n at least 25 in each of 3 independent experiments. (**J**) Based on lipidomic profiling at 6 hpf, the *sphk2^MZ* embryos are depleted of S1P (Sph-1P) and have elevated levels of sphingosine and (**K**) total long-chain ceramides compared to time-matched wildtype embryos. For J and K, results are the average of three independent experiments pooling 50 embryos each; errors bars indicate the standard error of the mean.

DOI: https://doi.org/10.7554/eLife.21992.003

and the deep cells of the blastoderm that will eventually form the embryo proper. During early gastrulation, the blastoderm and the YSL move toward the vegetal pole during a process known as epiboly. A microtubule network surrounds the yolk syncytial nuclei while another set of animal–vegetal-oriented microtubules emerge from the YSL into the yolk cytoplasmic layer (YCL). As epiboly progresses, two actin rings become visible; these rings serve to close the developing blastopore. Proper cytoskeletal patterning is crucial during epiboly as chemical disruption of either the microtubule or actin networks blocks epiboly progression (*Solnica-Krezel, 2006*).

To gain insight into why embryos treated with excessive sphingosine or with sphingosine kinase inhibitors fail to complete epiboly, we stained treated embryos to evaluate cytoskeletal markers associated with morphological defects. Compared to control embryos, those embryos treated with 5

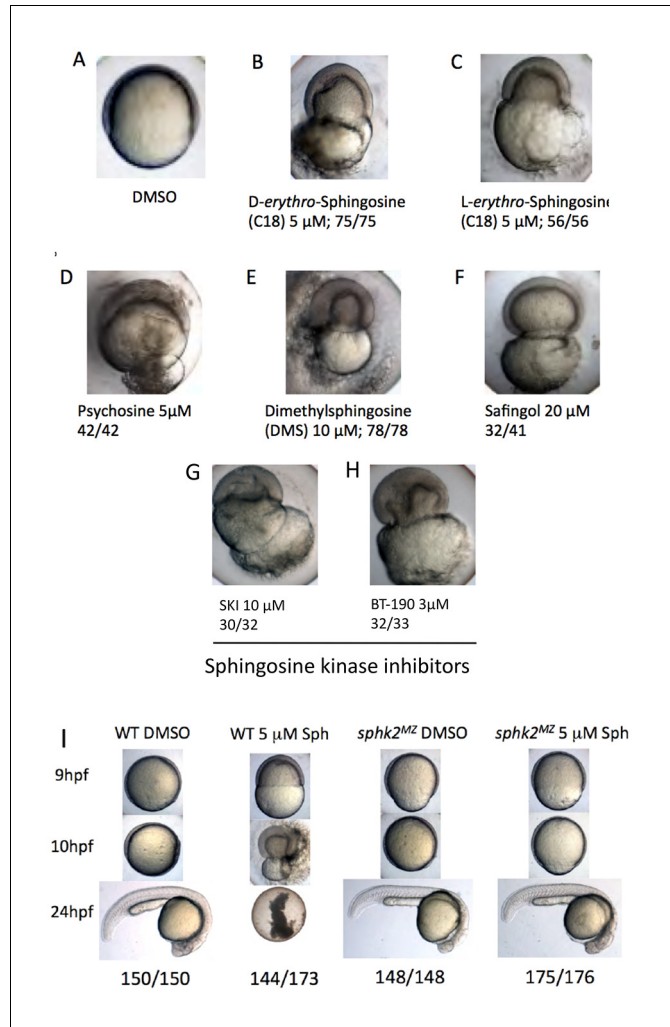

**Figure 2.** Excess sphingosine results in gastrulation defects and embryonic lethality, which is abrogated in the context of a *sphk2^MZ* mutation. (A) Embryos cultured in DMSO display normal coverage of the yolk by the developing blastoderm during epiboly at 7 hpf. (B) Treatment of wildtype embryos with 5 μM D-*erythro*-sphingosine (the active naturally occurring isomer) or (C) 5 μM L-*erythro*-sphingosine (the inactive isomer) caused stalling of embryonic development during epiboly, rupture of the yolk membrane, and failure of the embryos to complete gastrulation. This phenotype was recapitulated by embryos treated with (D) 5 μM D-galactosyl-ß1–1'-D-*erythro*-sphingosine (Psychosine), (E) 10 μM dimethylsphingosine (DMS), and (F) 20 μM L-*threo*-Dihydrosphingosine (Safingol). Treatment of wildtype embryos with two chemical inhibitors of sphingosine kinases, (G) SKI (10 μM) or (H) BT-190 (3 μM) recapitulated the catastrophic embryonic phenotype. (I) The majority (144/173, 83%) of wildtype (WT) embryos treated with 5 μM sphingosine exhibited a lethal phenotype during gastrulation, whereas the majority (175/176, 99%) of the *sphk2^MZ* embryos treated with 5 μM sphingosine completed gastrulaton and early development, only subsequently displaying the cardiac bifid phenotype observed in *sphk2^MZ* embryos, with no additional defects. Control embryos were cultured in DMSO vehicle (1%). Shown are representative embryos, n is indicated, from at least three independent experiments.

DOI: https://doi.org/10.7554/eLife.21992.004

μM sphingosine, 3 μM BT-190, or 10 μM SKI when stained with phalloidin show delayed progression with aggregation of thick F-actin positive bundles within the YCL (*Figure 3A–D*) as well as accumulation of F-actin within the cytoplasm of the EVL cells (*Figure 3E–H*). When control (DMSO-treated) embryos were at 50% epiboly, the treated embryos ranged typically from 25–40% epiboly. These thick actin bundles may cause abnormal contractile forces that disrupt the YCL; alternatively, these forces may create resistance to vegetal pulling of the EVL.

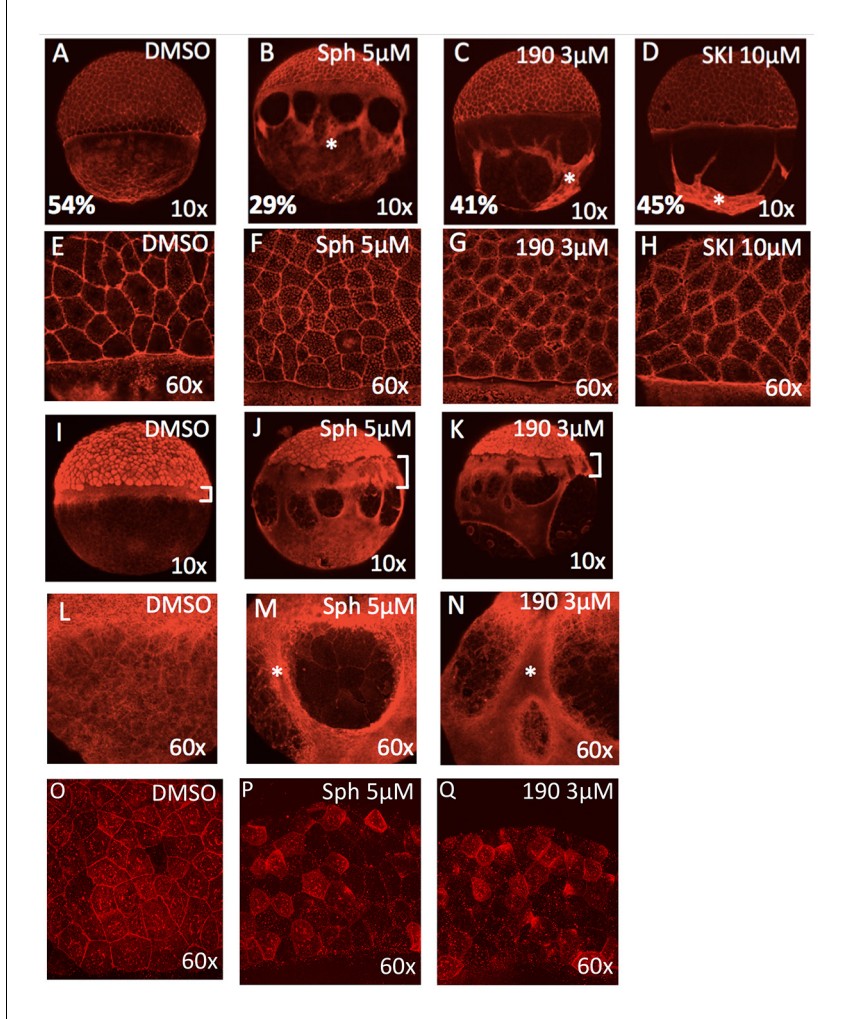

**Figure 3.** Excess sphingosine disrupts actin and microtubule organization and reduces junctional E-cadherin. (**A**) During gastrulation, actin accumulates at the vegetal pole as shown by phalloidin staining in DMSO-treated control embryos at ~50% epiboly (5.3 hpf). Phalloidin staining shows aggregation of thick F-actin positive bundles within the yolk cytoplasmic layer (YCL) of embryos treated with (**B**) 5 μM sphingosine, (**C**) 3 μM BT-190, or (**D**) 10 μM SKI (indicated by asterisks). For A-D, the average percent progression of the EVL is indicated as measured from the pole for a representative set of 5 embryos for each condition. (**E–H**) Shown at higher magnification, F-actin also accumulates within the cytoplasm of the enveloping cells in the treated embryos compared to DMSO-treated controls. During epiboly, two perpendicular microtubule layers are visible in DMSO-treated embryos at the YSL (**I**) and in the YCL (**L**). Staining with α-tubulin revealed a massively thickened microtubule layer within the YSL (as indicated by brackets, which also highlights the delayed progression of the EVL compared to deep layer cells) in embryos treated with (**J**) 5 μM sphingosine or (**K**) 3 μM BT-190 as well as (**M,N**) microtubule aggregation in certain areas of the YCL (denoted by asterisks) and other areas that are completely devoid of microtubules. Indicated by brackets in (**I-K**), the progression of the EVL appears to be more delayed compared to the deep cell layer. (**O**) DMSO-treated embryos display junctional E-cadherin staining of the EVL cells. (**P,Q**) Reduced junctional E-cadherin staining of the EVL cells and accumulation of staining within the EVL cytoplasm in embryos treated with 5 μM sphingosine or 3 μM BT-190. For all panels, representative embryos are shown, n is at least 100 from at least three independent experiments.

DOI: https://doi.org/10.7554/eLife.21992.005

Staining with an antibody against α-tubulin revealed a thickened microtubule layer within the YSL as well as microtubule aggregation in certain areas of the YCL and other areas that are completely devoid of microtubules (*Figure 3I–N*). Note that the progression of the EVL appears to be more delayed compared to the deep cell layer (indicated by brackets in panels I-K). E-cadherin staining

shows reduced localization at EVL cell membrane junctions and accumulation of staining within the EVL cytoplasm in embryos treated with 5 μM sphingosine or 3 μM BT-190, compared to DMSO-treated control embryos (*Figure 3O–Q*).

## Cers2b transcripts are enhanced in *sphk2^MZ^* embryos

The *sphk2^MZ^* embryos display elevated sphingosine and ceramide levels, yet are relatively resistant to gastrulation defects caused by excess sphingosine, suggesting that they employ a sphingosine kinase-independent mechanism to handle substrate accumulation. We hypothesized that *sphk2^MZ^* embryos might regulate one or more genes encoding enzymes responsible for the biosynthesis of sphingosine from ceramide (ceramidases) or conversion of sphingosine into ceramide (ceramide synthase). Therefore we examined expression levels for all the ceramide synthase and ceramidase genes that are annotated in the zebrafish genome (some of which are duplicated). This analysis showed only one gene, *cers2b*, that is significantly and strikingly upregulated in *sphk2^MZ^* embryos compared to wildtype embryos (*Figure 4A*). This suggests that *sphk2^MZ^* embryos up-regulate a salvage pathway for sphingosine turnover through a specific ceramide synthase gene, *cers2b*, which could potentially protect these embryos from sphingosine-mediated gastrulation defects.

Thus, we hypothesized that knockdown of Cers2b in *sphk2^MZ^* embryos should recapitulate the gastrulation phenotype observed following treatment of wildtype embryos with sphingosine. Knockdown experiments were performed using a translation blocking morpholino (MO) against *cers2b*. Because MOs can have off-target or toxicity effects, the MO was titrated to define reagent amounts, e.g. 5 ng, having no effect following injection into wildtype embryos. Injection of the same amount of *cers2b* MO into *sphk2^MZ^* embryos resulted in a phenocopy of the sphingosine-induced toxicity defect during early gastrulation in 60/288 (21%) embryos from four independent experiments, with failure of epiboly and pinching of the blastoderm from the yolk cell (*Figure 4B,C*, p=0.015). This suggests that transcriptional up-regulation of *cers2b* is important for *sphk2^MZ^* embryos to resist the buildup of excessive sphingosine and complete epiboly.

It is perhaps not surprising that the morphant phenotype is not fully penetrant since morphant knockdowns may be partial or transient. They may also be subject to off-targeting. Therefore, to more rigorously test whether Cers2b compensates for loss of Sphk2 to relieve excess sphingosine, we designed sgRNAs to target mutation of *cers2b*. We established a line of fish carrying a 23 bp deletion in exon6 of the *cers2b* gene that is predicted to delete most of the Lag domain and generate a frame-shift with a premature stop codon and a truncated protein that would lack enzymatic function. This mutant allele was tolerated for development and was crossed onto the background of the *sphk2* mutant, to raise *sphk2^MZ^*; *cers2b^+/-^* adult fish. When crossed to each other, the resulting clutches consistently had approximately 25% of the embryos that phenocopied the gastrulation defect caused by treating wildtype embryos with excess sphingosine (*Figure 4D*). These embryos rapidly perish, which precluded our ability to measure directly the ceramide levels or genotype the dying embryos. However, the genetic interaction of *sphk2* and *cers2b* is fully consistent with sphingosine-associated up-regulation and compensation from Cers2b. We note that the *cers2b* gene fails to be up-regulated in the *sphk2* morphants (*Figure 4E*), and neither do these embryos show increased ceramide levels (*Figure 4F*). Likewise, wildtype embryos treated with sphingosine fail to activate expression of *cers2b* (*Figure 4G*) or to significantly increase levels of total ceramides (*Figure 4H*). We crossed *sphk2^+/-^* females with either wildtype males or mutant null males, to compare the relative levels of *cers2b* transcripts or sphingosine. In the former case, the embryos are 50% heterozygous and 50% wildtype (none are null), while in the latter case 50% of the embryos are zygotic null (and the other 50% are heterozygous). If the *sphk2* zygotic null mutants were similar to *sphk2^MZ^* mutants, we would expect to measure half the levels of increased sphingosine in this latter cohort compared to what is measured in the *sphk2^MZ^* mutants. However, we found the levels were not increased compared to the control cohort and neither were overall S1P levels depleted (*Figure 4I*). Therefore, the data suggest that the increased sphingosine level associated with transcriptional activation of *cers2b* is dependent on the maternal loss of Sphk2 function, and neither wildtype embryos nor morphants are able to respond to increased sphingosine levels by up-regulation of the *cers2b* gene.

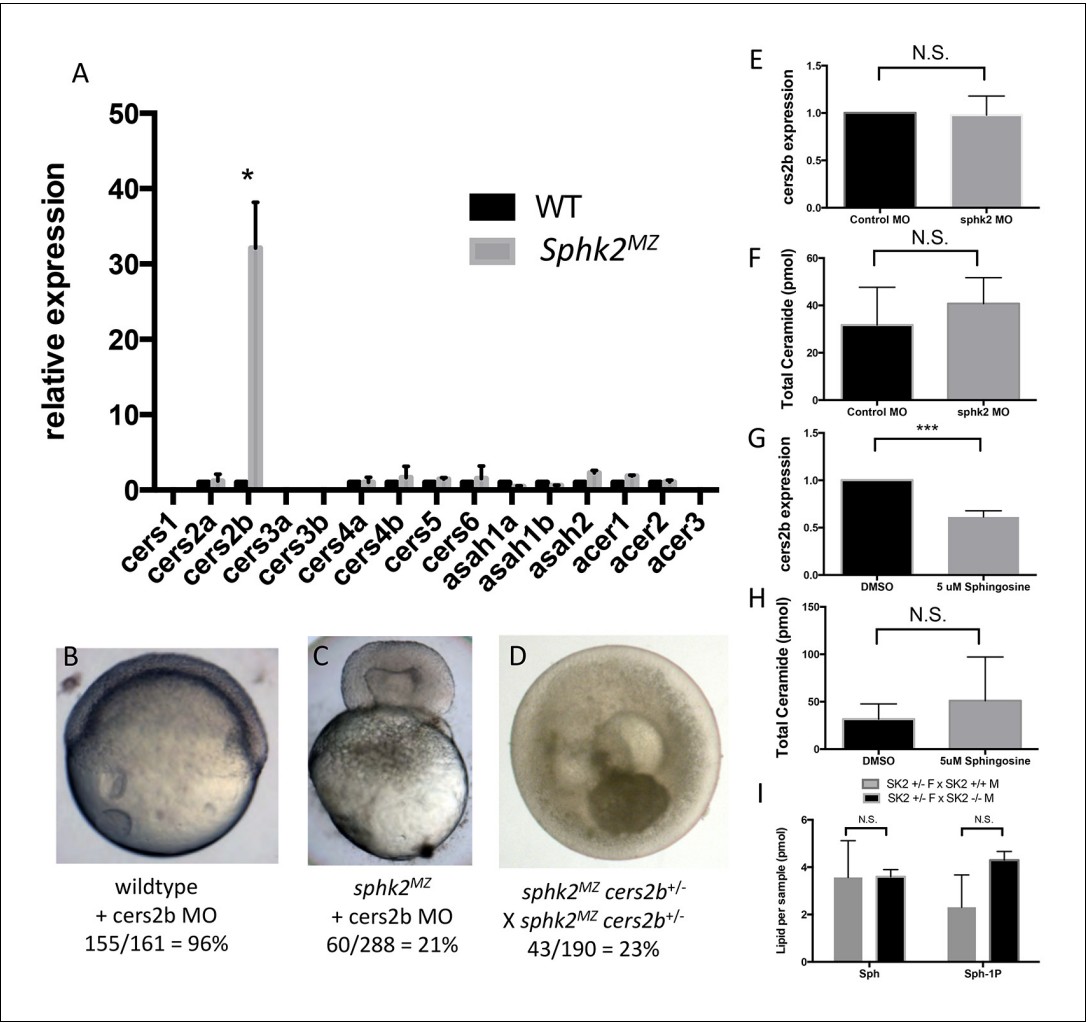

**Figure 4.** The *sphk2^MZ* embryos express strikingly high levels of *cers2b* transcripts. (**A**) Analysis of ceramide synthase and ceramidase genes by qRT-PCR shows that *cers2b* is significantly upregulated in *sphk2^MZ* embryos (p=0.019) compared to wildtype embryos at 2 hpf. (**B**) Representative wildtype or (**C**) *sphk2^MZ* embryos following injection with 5.3 ng of a *cers2b* translation blocking morpholino (MO), alongside (**D**) embryos derived from a *sphk2^MZ*; *cers2b^+/-* incross. For wildtype embryos, 155/161 (96%) completed normal gastrulation, epiboly, and appeared normal at 2 dpf. Following injection into *sphk2^MZ* embryos, 60/288 or 21% from 4 separate experiments recapitulated the early gastrulation lethality phenotype (p=0.0154). Correspondingly, 43/190 or 23% of embryos derived from crossing *sphk2^MZ*; *cers2b^+/-* adult fish recapitulated early gastrulation lethality phenotype by 9 hpf. (**E**) Sphk2 morphant embryos fail to upregulate *cers2b* transcripts and (**F**) did not vary in ceramide levels when compared to those injected with control morpholino. (**G**) Embryos exposed to 5 uM sphingosine in culture similarly fail to upregulate *cers2b* transcript levels and (**H**) showed no differences in ceramide accumulation. (**I**) There were no significant differences in sphingosine (Sph) or S1P (Sph-1P) levels in embryos with maternally deposited *Sphk2*, consistent with normal development. Embryos were harvested for both qPCR and lipid measurements at 6 hpf. All experiments were repeated at least three times with equivalent results.
DOI: https://doi.org/10.7554/eLife.21992.006

## Enhanced *cers2b* promoter activity in *Sphk2^MZ* embryos compared to wildtype embryos

The strikingly higher *cers2b* transcript levels in *sphk2^MZ* embryos suggest that this gene may have cis-regulatory elements that respond to sphingosine levels (sphingosine response elements). If true, these elements might confer promoter activity to a reporter gene, which like the endogenous gene should be more active in *sphk2^MZ* embryos compared to wildtype embryos. The *cers2b* gene is located on chromosome 16, with a transcriptional start site approximately 1.7 kb downstream of the

3' end of the adjacent *celf3b* gene. The putative promoter region of *cers2b*, including the entire 1680 bp non-coding region directly upstream of exon1 was isolated by PCR and cloned upstream of a firefly luciferase reporter gene. This construct was co-injected with RNA encoding a control *renilla* luciferase enzyme into fertilized eggs of wildtype or *sphk2^{MZ}* embryos. In three independent experiments using a total of 114 embryos of each genotype per experiment (n = 38 samples, each sample comprised from three embryos) the ratio between firefly luciferase to *renilla* luciferase (FL/RL) was 9-fold higher in *sphk2^{MZ}* embryos compared to wildtype embryos (*Figure 5A*, p<0.0001). We note that this difference was entirely attributable to the luciferase reporter, as the control RL levels were highly consistent in all experiments. To localize the sequences of the *cers2b* promoter that mediate this response, shorter promoter fragments were cloned into the reporter, progressively deleting 5'-upstream sequences, and compared for activity in wildtype and mutant embryos. The results of this study are also shown in *Figure 5A*. Deletion of the first 49 bp (1631 bp construct) decreased the enhanced responsiveness to 6-fold, while deletion of 83 bp (1597 bp construct) reduced it to a 2-fold difference. Deletion of the upsteam 400 bp (1281 bp construct) essentially eliminated promoter activity in both wildtype and mutant embryos. These results indicate that key regulatory regions conferring both promoter activity and enhanced response in *sphk2^{MZ}* mutant embryos are present in the 400 bp region at the 5' end of the 1.7 kb promoter construct. Indeed, when this 400 bp fragment was cloned upstream of an independent minimal SV40 promoter, although the basal activity was somewhat lower, the resulting construct fully recapitulates a 9-fold enhanced activity in *sphk2^{MZ}* embryos compared to wildtype (*Figures 5A* and 400 bp).

The ability of mutant embryos to enhance expression of *cers2b* in *sphk2^{MZ}* embryos indicates a sensing mechanism to engage this alternative metabolic pathway and resist toxicity caused by excess sphingosine. An attractive candidate component of this sensing mechanism would be Cers2b itself, since it is the ideal protein for recognizing sphingosine and thereby might be differentially impacted depending on sphingosine levels. We were struck by the additional observation that many vertebrate ceramide synthase enzymes, including Cers2b, encode not only metabolic activity domains, but also a homeobox domain, typically found in homeobox class transcription factors. This raises the possibility that Cers2b, in addition to functioning to sense sphingosine levels, could potentially impact the transcriptional activity of its own or other genes.

To test if Cers2b can impact expression of its own promoter, the luciferase reporter under control of *cers2b* upstream sequences was co-transfected into HEK293T cells with an expression vector for Cers2b or a control empty expression vector, along with a control *renilla* luciferase expression vector to normalize for transfection efficiency and lysate preparation. Expression of Cers2b consistently and significantly decreased expression of the *cers2b*:luciferase reporter (*Figure 5B*), indicating that Cer2b can be a functional repressor of its own gene. Addition of 10 µM sphingosine to the transfected cells relieved this repressor activity so that the reporter was expressed at levels not significantly different from the control conditions without Cers2b. To test if this activity was dependent on the homeobox domain, a two amino acid mutation was introduced into the Cers2b coding sequence (R121W, R122S), which is predicted to impair the conserved structure of the homeodomain (*Banerjee-Basu and Baxevanis, 2001*). Like addition of sphingosine, the RR121-122WS alteration eliminated the capacity for Cers2b to repress its own promoter (*Figure 5B*).

We also generated by site-directed mutagenesis an isoform for Cers2b that is mutant in the Lag domain (HH212-213DD). This isoform is predicted to be enzymatically dead, and so we compared the effect of this protein to the wildtype protein and the isoform with a mutant HOX domain for Cers2 promoter regulation. As shown *Figure 5B*, The Lag-mutant protein retains the ability to repress the reporter controlled by the *cers2b* promoter. Compared to the wildtype protein, the Lag-mutant fails to generate ceramide above background levels (*Figure 5C*). In contrast, the HOX mutant protein retains the ability, equivalent to the wildtype protein, to increase ceramide levels, but is unable to repress activity of the promoter. Therefore, the HOX domain is required for *cers2* gene regulation, independent of enzymatic function.

## Cers2b association with the nuclear membrane is enhanced by sphingosine exposure

Since forced expression of Cers2b can repress its own promoter, dependent on the HOX domain, we considered the possibility that Cers2b might be able to function directly as a transcription factor. In this case, it would be expected that some fraction of the protein is localized in the nucleus to

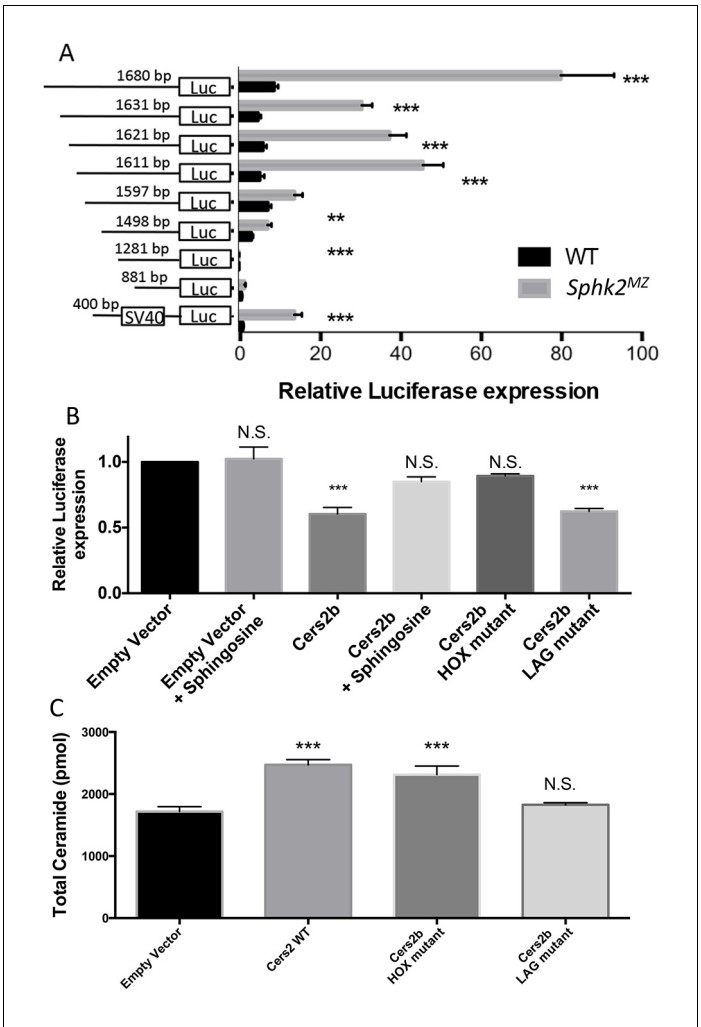

**Figure 5.** The *cers2b* promoter is relatively active in *Sphk2^MZ* embryos compared to wildtype embryos, and is repressed by Cers2b, dependent on the HOX domain. (**A**) Shown are relative luciferase activities from three independent experiments comparing the ratio between expression of firefly luciferase (FL) regulated by the *cers2b* promoter to *renilla* luciferase (RL), derived from co-injection of *renilla* luciferase RNA. The ratio (FL/RL on the Y axis) is significantly enhanced in *sphk2^MZ* embryos compared to wildtype embryos, p<0.0001. To demonstrate that this reflects cis-regions of the *cers2b* promoter, shorter promoter fragments containing progressively less upstream sequences were similarly compared. Much of the promoter activity was found to reside in the upstream 400 bp, and transfer of this sequence to a heterologous (minimal SV40) promoter was sufficient to transfer the response in *sphk2^MZ* embryos. (**B**) Similar luciferase reporter assays were carried out in HEK293T mammalian cells. Ectopic expression of Cers2b caused a significant repression of the *cers2b* promoter-regulated reporter compared to empty expression vector control. This repression was relieved in cells treated with 10 mM sphingosine, which does not affect basal level of the reporter (empty vector +sphingosine). Repression was also abrogated by mutations predicted to disrupt function of the homeobox (HOX) domain, while a mutation in the catalytic (LAG) domain did not impact the capacity of the protein for repression. (**C**) When Cers2b is overexpressed in HEK293T cells, the ceramide levels increase due to its catalytic activity, which is unaffected by the homeobox (HOX) mutation but disrupted entirely by the catalytic (LAG) domain mutation. This indicates the repressive activity on the promoter is independent of the catalytic activity of Cers2. All results are averaged from three independent experiments; bars indicate standard error of the mean. Statistical significance is derived by one-way ANOVA and indicated by * (p<0.05), ** (p<0.01), *** (p<0.001). N.S. denotes not significant.
DOI: https://doi.org/10.7554/eLife.21992.007

interact with its own promoter or other genes. We used confocal microscopy of HEK293T cells transfected with a vector to express a Cers2-GFP fusion protein, in order to visualize localization in live cells (*Figure 6A*). As expected, Cers2-GFP protein is localized primarily to the ER, throughout the cell including contiguous to the nuclear membrane. Following exposure to 10 µM sphingosine, a condition that leads to de-repression of the reporter that is regulated by the *cers2b* promoter, there is a considerable change in the distribution of the fusion protein and notably enhanced signal associated with the nuclear membrane. There did not appear to be significant levels of the fusion protein in the nucleus either at steady state or following sphingosine exposure. We also compared expression levels of the protein by western blotting, following separation of nuclear and cytosolic extracts. In this case there was a marked enrichment for Cers2b in the nuclear fraction following exposure to sphingosine (*Figure 6B*). However, when the nuclear fraction was further separated, we found that sphingosine induction caused Cers2b association primarily with nuclear membranes (*Figure 6C*), consistent with the confocal imaging. Therefore, we did not find evidence that Cers2 functions as a nuclear protein and while function as a transcription factor can't be ruled out, it appears more likely that Cers2b impacts promoter activity by an indirect mechanism, perhaps acting at the nuclear membrane to process nuclear sphingosine.

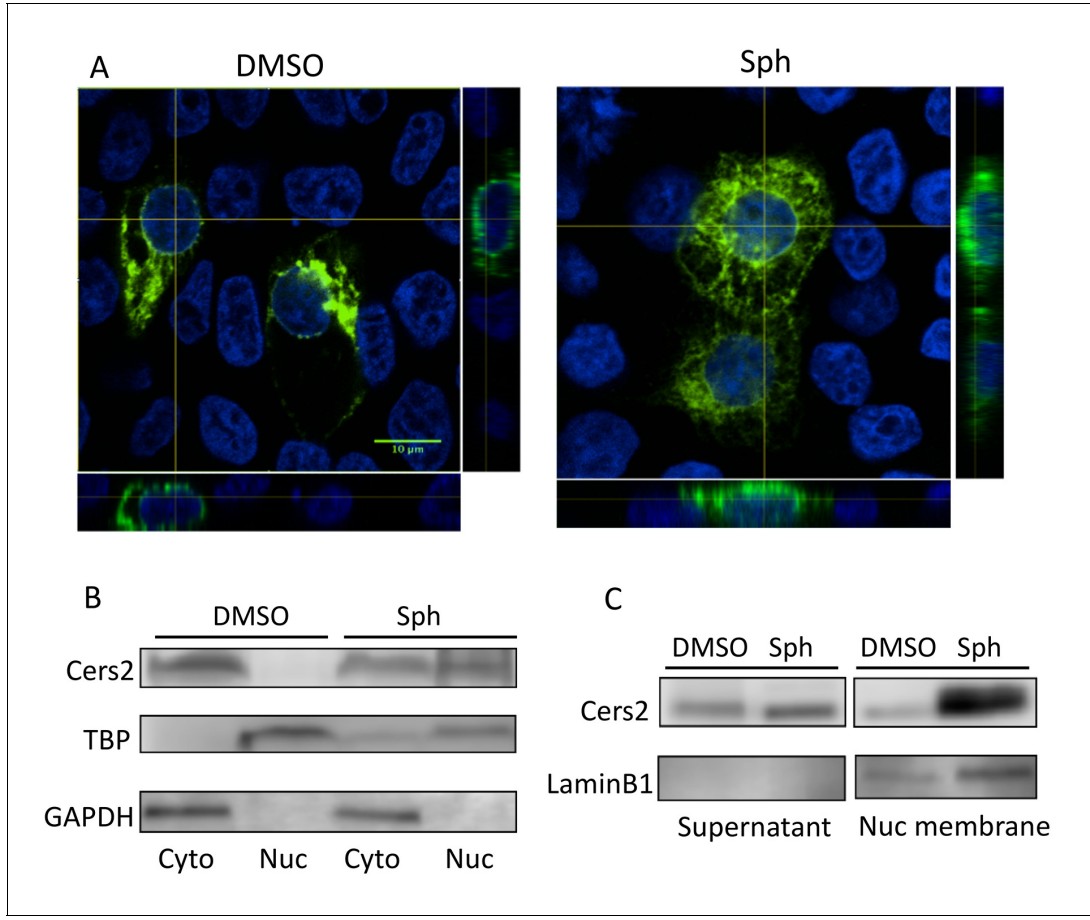

**Figure 6.** The subcellular localization of Cers2 protein is altered by increased sphingosine levels. HEK293T cells transfected with a Cers2-GFP expression plasmid were examined after 6 hr exposure to 10 uM sphingosine in culture. (**A**) Cells treated with sphingosine (Sph) show subcellular relocalization of Cers2b-GFP (green) with more overlap within nuclear regions (DAPI, blue) as seen in the z-stack (side panels). (**B–C**) Western blotting analysis of lysates probed with antibodies specific to Cers2, GAPDH (as a control for cytoplasmic proteins), TBP (as a control for nuclear proteins) or LaminB1 (as a control for nuclear membrane associated proteins). (**B**) Subcellular fractionation into cytoplasmic (Cyto) and nuclear (Nuc) lysates confirms increased nuclear association of Cers2 in cells treated with sphingosine. (**C**) Fractionation using a nuclear membrane isolation kit indicates Cers2 is most enriched in the nuclear membrane fraction.

DOI: https://doi.org/10.7554/eLife.21992.008

## The *cers2b* gene is activated during oogenesis in the absence of Sphk2, associated with increased sphingosine levels

Given the early activation of Cers2b in the *sphk2$^{MZ}$* embryos, we hypothesized that this could represent a protective mechanism to remove excess sphingosine during oogenesis. In this case, we would expect that sphingosine levels would be relatively high and the *cers2b* gene already activated even in mutant oocytes. Lipidomics comparing wildtype and *sphk2* mutant oocytes confirmed, similar to the *sphk2$^{MZ}$* embryos, a lack of S1P and relatively higher levels of sphingosine in the mutant oocytes (*Figure 7A*), although in this case the total ceramide levels were not significantly different (*Figure 7B*). We also carried out in situ hybridization experiments probing oocytes derived from ovaries of *sphk2* mutant females. This confirmed strikingly higher levels of *cers2b* transcripts in the

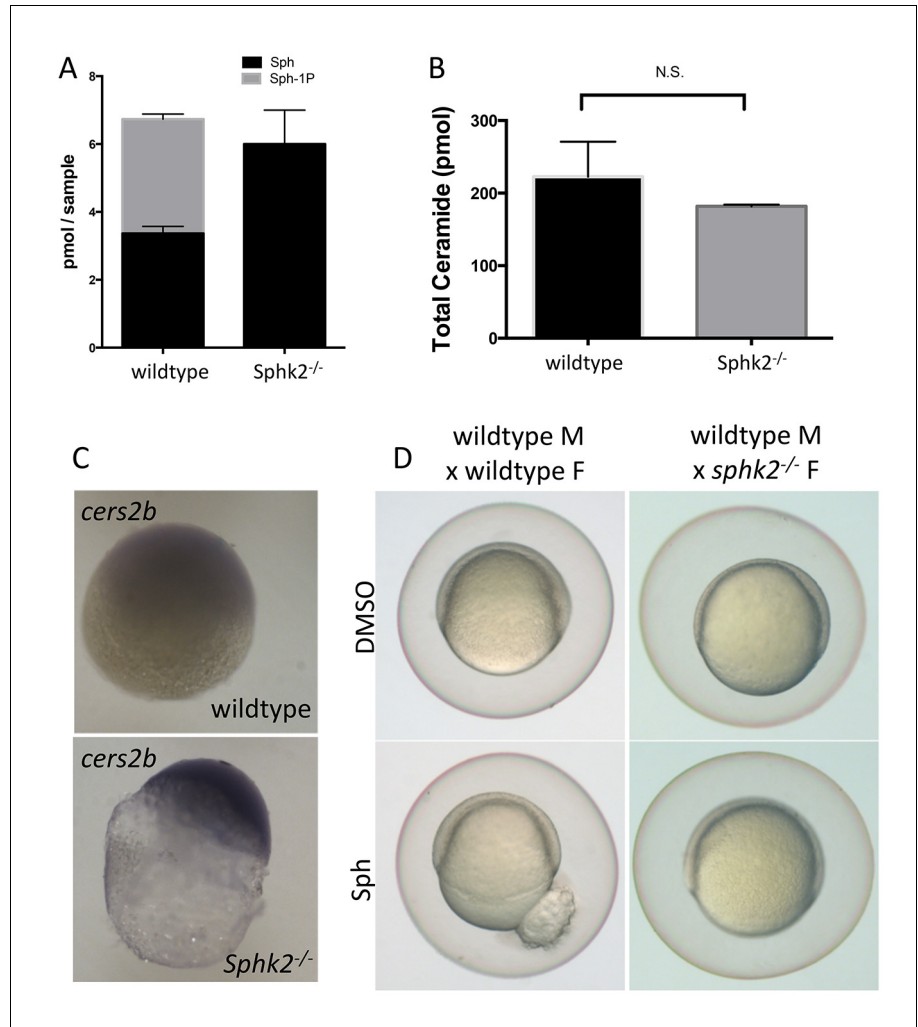

**Figure 7.** Transcriptional changes during oogenesis underlie *sphk2$^{MZ}$* resistance to sphingosine buildup. Oocytes isolated from wildtype and *sphk2$^{-/-}$* female fish show (**A**) increased sphingosine (Sph) levels in mutant oocytes associated with loss of S1P (Sph-1P) with (**B**) no difference in total ceramide levels. (**C**) Oocytes derived from *sphk2$^{-/-}$* oocytes showed significant upregulation of *cers2b* transcripts, as imaged following in situ hybridization. Note that signal in the wildtype oocytes was equivalent to background staining using a sense-strand control probe. (**D**) Correspondingly, embryos derived from *sphk2$^{-/-}$* females (right) showed relative tolerance to sphingosine accumulation when compared to those from wildtype females (left) when cultured in medium with 5 uM sphingosine. For (**A, B**) . results are averaged from three independent experiments using at least 50 embryos per measurement. For (**C,D**), representative embryos are shown, n > 25, from at least three independent experiments.
DOI: https://doi.org/10.7554/eLife.21992.009

mutant oocytes compared to wildtype (*Figure 7C*). Furthermore, early embryos derived from a *sphk2* null female are resistant to rupture when exposed to exogenous sphingosine, despite a functional *sphk2* allele inherited from the wildtype male (*Figure 7D*). All the data are fully consistent with a sphingosine-induced Cers2b-based protective mechanism during oogenesis.

## Discussion

S1P signaling has great capacity to impact embryonic development through complex receptor-dependent G-protein coupled pathways that cross-talk with each other and additional developmental signaling programs. It is less appreciated that regulation of sphingolipid metabolism may have additional receptor-independent functions equally critical for early developmental decisions such as cell proliferation, survival, and tissue morphogenesis. Previous studies confirmed that Sphk2 is an enzyme required to generate S1P used during early embryogenesis in an S1P receptor-dependent manner to regulate endoderm convergence and heart tube formation (*Hisano et al., 2015a*; *Mendelson et al., 2015*). Here we confirmed that $sphk2^{MZ}$ mutant embryos are depleted of S1P, but we also found additional previously unrecognized phenotypes that result from dysregulation of sphingolipid metabolism. Importantly, the mechanistic underpinnings of this phenotype revealed a previously unknown regulatory mechanism by which sphingolipid metabolism is achieved at the level of ceramide production.

The $sphk2^{MZ}$ mutant embryos have increased sphingosine levels due to the failure to convert this substrate to S1P. These embryos are developmentally delayed during gastrulation, although they eventually recover and can complete epiboly. This delay can be phenocopied in wildtype embryos by culturing in the presence of excess sphingosine, suggesting that the two observations are linked. Even with the increased sphingosine levels, the mutant embryos are nevertheless relatively resistant to toxicity caused by excess sphingosine. This indicates that they have engaged an alternative program to convert sphingosine to a non-toxic metabolite. The mutant embryos express strikingly high levels of *cers2b* transcripts, strong evidence for this anticipated salvage pathway for sphingosine turnover. Knockdown and genetic interaction experiments suggest that transcriptional up-regulation of *cers2b* is important for mutant embryos to complete epiboly. Finally, *cers2b* promoter activity can be transferred to an independent reporter gene, functioning in cis to promote higher expression levels in $sphk2^{MZ}$ embryos compared to wildtype embryos. Upstream sequences confer sphingosine responsivity, located within a 400 bp region. Cers2b protein itself can partially repress activity of this promoter, dependent on a HOX domain but independent of enzymatic activity, and this repression is relieved by addition of sphingosine. These findings reveal a previously unrecognized mechanism by which the genome senses and modulates sphingosine levels during embryogenesis.

The $Sphk2^{MZ}$ embryos are delayed in epiboly is not known, but we speculate that it is caused by destabilization of the actin-microtubule cytoskeletal network due to accumulation of excess sphingosine. A similar delay is recapitulated in wildtype embryos treated with 3 µM sphingosine, while increasing this to 5 µM causes rupture of the epiblast associated with massive disorganization of actin and microtubule networks that are essential for 'pulling' the blastoderm around the yolk cell. Upregulation of *cers2b* in the mutant embryos manages to maintain sphingosine levels below this 'catastrophic' threshold, generating a phenotype similar to wildtype embryos treated with the 3 µM lower dose. A good candidate for mediating the connection to cytoskeleton is the p21-activated kinase Pak1. Pak kinases can localize to cortical actin structures, and expression of activated mutants induces cytoskeletal reorganization, including loss of actin stress fibers (*Manser et al., 1997*). Pak1 signaling is also implicated during epiboly in sustaining E-cadherin at adherens junctions (*Tay et al., 2010*), which also fails in the sphingosine-treated embryos. Interestingly, sphingosine was shown previously to activate PAK1 in mammalian cells in a concentration-dependent manner (*Bokoch et al., 1998*). The dramatic blastoderm constriction phenotype is remarkably similar to the maternal-zygotic *betty boop* (*bbp*) mutant embryo (*Holloway et al., 2009*), suggesting that it could also involve defects in a p38/MAPKAPK2 pathway. Although S1P$_2$ couples through G$\alpha_{13}$ to control cell migration (*Ye and Lin, 2013*), and G$\alpha_{13}$ signaling plays an important role in epiboly progression (*Solnica-Krezel, 2006*), the receptor-dependent pathway seems less likely to be involved, since similar phenotypes are not observed in embryos lacking S1P receptors (*Hisano et al., 2015b*).

On the other hand, it has long been recognized that while Sphk1 is localized primarily in the cytoplasm, Sphk2 is mostly a nuclear protein, and may be particularly important for regulating nuclear sphingosine levels, which could impact DNA synthesis (*Igarashi et al., 2003*) or oxidative stress otherwise leading to DNA damage and apoptosis (*Van Brocklyn and Williams, 2012*). We note that *sphk2* is expressed as a maternal transcript, and this is also true of *cers2b* in the *sphk2^{MZ}* mutant embryos. This suggests that the genome might be especially vigilant at limiting nuclear sphingosine levels during oogenesis in order to protect genomic integrity. In this context, it is of interest to note previous studies where sphingolipids, particularly S1P, were shown to protect oocytes from apoptosis induced by genotoxic stress or chemotherapy in mice and primates (*Morita et al., 2000*). S1P treatment in vitro was shown to induce oocyte survival (*Hannoun et al., 2010*), and acid ceramidase is known to modulate oocyte apoptosis (*Eliyahu et al., 2010*). Thus, our findings could have practical implications in reproductive sciences and in vitro fertilization technology.

It is interesting that Cers2b contains a homeobox domain, required for repressing transcriptional activity of its own promoter. This leads to a model whereby under conditions with limiting sphingosine levels the Cers2b enzyme functions to repress its own gene and thereby limit sphingosine conversion to ceramide (perhaps also favoring conversion to S1P). If sphingosine levels exceed a certain threshold, for example by loss of Sphk2 activity and/or increased ceramidase activity, this is recognized by binding of sphingosine to the Lag domain of Cers2b, which relieves Cers2b repression activity, perhaps through a conformational alteration, and activates enzymatic activity to remove sphingosine by metabolism to ceramide. Currently we can't rule out post-transcriptional regulation of the *cers2b* gene in response to increased sphingosine levels. However, we note that the *cers2b* transcript levels are markedly enhanced, at levels that are recapitulated by the promoter using a luciferase reporter with no shared RNA sequences. Furthermore, the increase is highly specific to the *cers2b* transcript.

15 years ago Venkataraman and Futerman speculated whether the presence of a homeobox domain suggests that the enzymes might act as dual sphingosine sensors and transcriptional regulators, analogous to how sterol regulatory element binding protein (SREBP) senses and modulates sterols (*Venkataraman and Futerman, 2002*). Consistent with our results, overexpressing mammalian Cers2 in HT29 colon cancer cells caused a modest but significant reduction in transcription of a reporter gene regulated by an acid ceramidase promoter (*Tirodkar et al., 2015*), suggesting that Cers2 might have additional pathway-relevant targets. The single *Drosophila* Cers ortholog, Schlank, also has a homeodomain, in addition to the Lag motif. Interestingly, *schlank* mutants could be rescued by expression of variant proteins lacking a functional Lag motif, as long as they contained the homeodomain and a nuclear localization signal (*Voelzmann et al., 2016*). While the mechanism for Cers2b action during zebrafish embryogenesis remains unclear, our results suggest the critical role played by this ER/nuclear membrane-localized enzyme is in autoregulation of sphingolipid metabolism.

Our data provide new evidence for a role of ceramide synthase in vertebrate embryonic development, but the concept of a dual role in metabolism and transcription has been considered previously. In fact, together with our observations, it seems likely that Cers2b may play a central role in coordinating a response following sensing of sphingosine levels to the transcriptional machinery in the regulation of sphingolipid metabolism.

Our observation linking Cers2b to genomic sensing/transcription was only made possible by close examination of subtle phenotypes in the *sphk2^{MZ}* mutant embryo. The identical phenotype caused by excess sphingosine in wildtype embryos (pinching off of the blastoderm during epiboly and embryonic lethality) was also seen in *sphk2* morphants. This suggests that mutants (but not morphants or wildtype embryos treated with sphingosine) are able to recognize the defect and have time to take corrective action through engaging the salvage Cers2 pathway. The process apparently occurs during oogenesis, since it involves the expression of maternal transcripts. The capacity of zebrafish embryos to compensate for mutant genes may be relatively robust (see also [*Rossi et al., 2015*]), and at least partially responsible for the reported disparity between mutant and morphant phenotypes (*Kok et al., 2015*). It can also be fortuitous for revealing novel biological regulatory mechanisms.

## Materials and methods

### Maintenance of zebrafish

Wildtype (AB/TU hybrid) and mutant zebrafish were maintained at 28.5C and staged as described (*Kimmel et al., 1995*). $Sphk2^{MZ}$ were generated ($sphk2^{wcm2}$) and characterized as described previously (*Mendelson et al., 2015*). To generate a *cers2b* mutant line, a sgRNA was designed using CHOPCHOP (https://chopchop.rc.fas.harvard.edu) to target the 20 bp sequence AGTACCAGTAC TGAGACGGC in exon6 within the Lag domain of the *cers2b* gene. The sgRNA was synthesized using 125 ng gblock (IDT) and transcribed in vitro using the MegaShortScript T7 kit (ThermoFisher, Waltham, MA). 150 pg RNA was injected into fertilized eggs with 250 pg recombinant Cas9 protein (Pnabio) in a volume of 2 nl per injection. F0 founder fish transmitting mutant alleles were identified using T7 exonuclease assays and F1 embryos raised to adulthood. One line was established ($cers2b^{wcm18}$) that transmits a 23 bp deletion allele, with loss of the nucleotide sequence GTCTCAGTACTGGTACTACATGC, causing a frameshift that deletes Cers2b starting at amino acid S178, generating a mutant protein encoding subsequently:

YDLEEMWKGFPTLTLLPAGTGLLHLSPVQCGVGRQAQGL*.

### Quantitative RT-PCR

Staged embryos were homogenized with TRIzol (Ambion) and total RNA was isolated (Qiagen). Total RNA (1 µg) was used to generate cDNA using reverse transcriptase and random hexamers (Roche). LightCycler 480 SYBR Green 1 Master Mix (Roche) was used to analyze cDNA by quantitative RT-PCR using the Light Cycler 480II (Roche). The PCR cycle conditions were 95C for 15 min followed by 40 cycles at 94C for 14 s, 54°C for 30 s, and 72°C for 30 s. $C_t$ values were calculated using the $\Delta\Delta C_t$ method (*Livak and Schmittgen, 2001*), based on the median value from a triplicate set. Each value was normalized to levels of *18S* transcripts. Statistical significance was determined using a two-tailed Student's t-test. Primer sequences were designed using Primer3 and were:

cers1 FOR: 5'-ACATGGACGAATGGAGGAAG
REV: 5'-CGAGAATGCCTATGTTGTGG
cers2a FOR: 5'-AACCAAGAGCGACCAAACC
REV: 5'-CCAGCAATGAAAGCAACAAG
cers2b FOR: 5'-CGCAGAAATCAAGACAGACC
REV: 5'-AGCAAGACCACCGATGAAG
cers3a FOR: 5'-TCAGGAGGAGACGAAACCAG
REV: 5'-CCTCCAATGAATGCCAACAG
cers3b FOR: 5'-TGTAAAAAGCTGGGCTGGTC
REV: 5'-CTCCTCCAAAAGTCGAGCAC
cers4a FOR: 5'-CAAACTGGAGGCGTTCTACC
REV: 5'-AGCCATGACTGAATCTGACG
cers4b FOR: 5'-GAAACCGCAGAAATCTGGAC
REV: 5'-ACCAGGTAGAAGGCAAACCTC
cers5 FOR: 5'-ACAAACCCAGCACAAGAACC
REV: 5'-CAGAAAACGCATCCCATACG
cers6 FOR: 5'-AATCCAACGCTGGTTCAGAC
REV: 5'-TTCTTCAGGAAGCGCACAC
asah1a FOR: 5'-TATGTTGGCATGCTCACTGG
REV: 5'-CCCATCAAAGTCAAAGCGTTC
asah1b: FOR: 5'-CATGATTCAGGCCATCAGAG
REV: 5'-GGCAGAGTGTCCACCATTAAC
asah2: FOR: 5'-TAAGAGAGTCGTGTTCGTCACC
REV: 5'-TCCTGAATGTGTGTGTGTGC
acer1 FOR: 5'-CAAACACTTCCCTTCCTTCG
REV: 5'AGGCATTGGCTGTAGGTTTG
acer2 FOR: 5'-GTGGTTTCCCAAGAGATACCTG
REV: 5'-TGGAGTTTATCGCTGGCTTG
acer3 FOR: 5'-ATCTACAGCTGCTGCGTCTTTG

REV: 5'-AATAGTTAACGGCACGCTCTTG

## Chemical treatment of zebrafish embryos

Wildtype and *sphk2^MZ^* zebrafish embryos were obtained at the one cell stage from paired adults, cultured in system water until 2 hpf and then grouped into individual 60 mm petri dishes, each containing 35 embryos in 10 mL of 1x E3 buffer, 1% DMSO, and 1 mM Tris pH 7.5. Compounds were diluted in DMSO to 10 mM stocks, and added to the cultured embryos to a final concentration of: 5 µM D-*erythro*-sphingosine C18 chain (Avanti), 5 µM L-*erythro*-sphingosine C18 chain (Matreya), 5 µM D-galactosyl-ß1–1'-D-*erythro*-sphingosine (Psychosine) (Avanti), 10 µM dimethylsphingosine (DMS) (Tocris), 20 µM L-*threo*-Dihydrosphingosine (Safingol) (Avanti), 20 µM C8 ceramide (Enzo Life Sciences), 20 µM spingosine-1-phosphate (d18:1) (S1P) (Avanti), 20 µM N-acetyl-D-sphingosine (Sigma), 10 µM sphingosine kinase inhibitor 2 (SKI, Cayman Chemicals), or 3 µM BT-190. BT-190 compound is identical to RB-046 and was synthesized as described (*Baek et al., 2013*). For these experiments the shorter chain ceramides were used because they can be dissolved and accurately measured. In contrast, long-chain ceramides are extraordinarily difficult to manipulate in vitro due to hydrophobicity making it difficult to accurately define the exposure level of phospholipid. The morphology of developing embryos was scored at 6 hpf.

## Morpholino oligomer injection

A translation blocking morpholino oligomer (MO) was designed to target the 5' UTR around the start codon of Cers2b to block mRNA translation (5'-CTCGCTCAGACCCGCCAGCATTTCA) and was purchased from Gene Tools (Philomath, OR). Blast analysis indicated the MO is specific for *cers2b* (no overlap with other sequences). All morphants were compared to stage matched embryos that were injected with the same concentrations of a standard control morpholino (Gene Tools). Each MO was titrated by injection into 1–4 cell fertilized embryos to determine a minimal dose for the reproducible phenotype. Neither the control or *cers2b*-specific MO disrupted normal development of wildtype embryos. Microinjection of MOs was performed using a PLI-100 Pico-Injector (Harvard Apparatus).

## Preparation of tissue samples for lipid analysis

Fifty zebrafish embryos were dechorionated using protease for 5 min and washed using E3 solution. The pellets were centrifuged for 2 min and the supernatants removed. The samples were submitted to the analytical core facilities at Stonybrook University or the Medical University of South Carolina for ceramide and sphingolipid analysis. Sphingolipids were extracted after the addition of internal standards and quantified by LC/MS/MS as described previously (*Mendelson et al., 2013*).

## Immunofluorescence for F-actin staining

Embryos were fixed overnight in 4% paraformaldehyde in PBS at 4C. Following two washes for 5 min each in PBSTw (PBS + 0.1% Tween-20), embryos were manually dechorionated. To permeabilize, embryos were incubated for 2 hr in PBS + 2% Triton X100. Following two washes for 5 min each in PBSTw, embryos were blocked for 1 hr with 2% BSA in PBSTw. Embryos were incubated for 2 hr at RT in Alexa 546 phalloidin (Invitrogen) diluted 1:40 in 2%BSA/PBSTw. Following 6 washes for 10 min each in PBSTw, embryos were mounted in agarose for confocal microscopy.

## Immunofluorescence for microtubule staining

Following fixation in microtubule stabilizing buffer [MSB] at RT for 2 hr, embryos were rinsed briefly in PBSTw and dechorionated. MSB composition is 80 mM KPIPES (pH6.5), 5 mM EGTA, 1 mM MgCl2, 3.7% formalin, 0.25% gluteraldehyde, and 0.2% Triton. Embryos were blocked in 10% Normal Goat Serum (Vector Labs) in PBSTw for 30 min and then incubated with monoclonal anti-α-tubulin (1:500) (clone DM1A) diluted in blocking reagent overnight at 4C. Following four washes in PBSTw, the embryos were incubated in Alexa 568 goat anti mouse IgG (Invitrogen) (1:500) diluted in blocking reagent for 2 hr at RT and then rinsed four times for 20 min each in PBSTw prior to mounting in agarose and confocal imaging.

## Immunofluorescence for E-Cadherin

Embryos were fixed with 4% PFA in PEMTT (0.1 M PIPES, 5 mM EGTA, 2 mM MgCl$_2$ · 6H$_2$O, 0.1% TritonX-100, 0.1% Tween 20, pH 6.8) overnight. Embryos were then washed with 1 × PEMTT buffer and manually dechorionated. Embryos were incubated for one hr with blocking solution (1% DMSO, 10% Normal Goat Serum, 200 μM KCl in PEMTT) at RT. The embryos were incubated with mouse anti-E-Cadherin antibody (BD) in blocking solution overnight at 4C, which was followed by washing with 1 × PEMTT and incubation with Alexa 568 goat anti mouse IgG (Invitrogen) (1:500) diluted in blocking solution for 2 hr. The embryos were finally washed with PEMTT and then mounted for imaging.

## Imaging analysis

Fish were anesthetized using Tricaine (United States Biochemical) prior to imaging. Brightfield images were taken using a Nikon SMZ1500 fluorescence microscope with an Insight Firewire two digital camera and SPOT advanced software. Fluorescent images were taken using a Zeiss Axio Observer.Z1 microscope and captured using a Zeiss AxioCam CCD camera. For confocal analysis, fixed and stained embryos were mounted in 1% low melt agarose (National Diagnostics) dissolved in water. For cell imaging, 10,000 HEK 293 T cells were plated onto four well MilliCell EZ slides (EMD Millipore) and transfected as described below. Cells were fixed in 4% paraformaldehyde for 5 min, washed three times in PBS, permeabilized with 50 ug/mL digitonin in PBS for 5 min at 37C, then quenched with 50 mM NH4Cl at 37C for 5 min. After 3 washes for 5 min each in PBS, cells were blocked in 5% goat serum at 37C for 30 min and washed for 3 times for 5 min each in PBS. DAPI was added at 1:2000 dilution by volume for 2 min, then washed for 5 times for 5 min each in PBS. Slides were mounted in ProLong Gold Antifade reagent (ThermoFisher Scientific, Waltham, MA) overnight at 4C. Confocal images were taken using an Olympus Fluoview Microscope with a 60x lens and analyzed using Fluoview software. Images were processed with ImageJ (1.42q) Imaris (Bitplane Inc.) and Adobe PhotoshopCS4 software.

## Reporter plasmids and deletion constructs

A genomic region including 1680 bp of sequence upstream of the first exon of *cers2b* was cloned into the TOPO TA vector and transferred to the Kpn1/Xho1 sites of the pGL3 basic firefly luciferase reporter plasmid (Promega). To generate shorter promoter constructs the same 3' primer was used along with 5' primers targeting progressively proximal sequences. These PCR fragments were cloned into TOPO TA and transferred to the Kpn1/Xho1 sites of the pGL3 basic firefly luciferase reporter plasmid. A construct containing the first 400 bp of sequence (the most distal 5' sequences of the full-length promoter construct) was also generated by PCR and cloned into a pGL3 promoter firefly luciferase reporter plasmid containing an SV40 promoter (Promega). Primer sequences are:

 1680 bp: 5'-ATGAAAATATATGTGAATAAACTGCAA

 1631 bp: 5'-AAACGAAGGTTGAGGGAACG

 1621 bp: 5'-TGAGGGAACGTTTCTTCATG

 1611 bp: 5'-TTTCTTCATGGTTTAAAAGCGTTC

 1597 bp: 5'-AAAAGCGTTCTTTAGATTATTAAAATG

 1498 bp: 5'-AAAACTGAATATTGAGTAGGGGG

 1281 bp: 5'-AAACTCAGTCTCCCGCAAAGTT

 881 bp: 5'-TAGGCTATGATGTTGGGTTACTTTGT

 common reverse primer: 5'-GTAACTCACGTAGGATCGGC

Two point mutations in the HOX domain (R121W, R122S) were generated using the full-length 1680 bp pGL3 construct via site directed mutagenesis. Likewise, two point mutations in the Lag domain (H212D, H213D) were similarly generated. Mutations were confirmed by directly sequencing the plasmids.

## In vitro transcription

RNA used for micro-injection was obtained by in vitro transcription using the linearized vector pRL-SV40 (Promega). Following linearization of the template with BamH1, one microgram was used to generate capped mRNAs with the mMessage mMachine kit (Ambion), followed by precipitation with LiCl. RNA was quantified by optical density, and integrity was confirmed by gel electrophoresis.

## Microinjection and luciferase assays

Single cell staged embryos were injected with 20 pg of luciferase reporter DNA and 0.05 pg of RNA encoding Renilla luciferase (to control for injection, lysate harvest, etc.). Following microinjection, embryos were cultured in 1x E3 solution until 50% epiboly, at which point three embryos were collected and lysed in 50 µl of 1x Passive Lysis Buffer (Promega). Luciferase assays were performed according to the manufacturer's instructions. For each assay, 20 µl of lysate was added to 100 µl of luciferase substrate [LARII], and luminosity was measured in a Turner TD-20e luminometer. The signal was quenched using 100 µl of stop and glo buffer and renilla luciferase activity was measured. The firefly luciferase values were divided by corresponding values of the control renilla luciferase.

## Transfection assays

Human (embryonic kidney) 293T/17 cells were purchased from ATCC (CRL-11268), with validated certificate of analysis provided by ATCC. Cell cultures were routinely tested and found negative for mycoplasma. 250,000 293 T cells per well in 24 well dishes were transfected with Lipofectamine LTX Reagent with Plus Reagent (ThermoFisher, Waltham, MA) with 250 ng of a PCS2 +expression vector, with or without the *cers2b* cDNA, along with 250 ng of pGL3 reporter plasmid as described above and 5 ng of CMV-RL to normalize for transfection efficiency and lysate harvest, for 24 hr. Cells were then washed with PBS and incubated with or without 10 µM sphingosine for 6 hr. Cells were subsequently lysed in 100 µl of 1x Passive Lysis Buffer per well (Promega). Luciferase assays were performed as described above.

## Cell lysates and western blotting

293 T cells were cultured in 10 cm dishes to 70% confluence and transfected with 6 µg of Cers2 human ORF expression plasmid (OriGene) using Lipofectamine LTX Reagent with Plus Reagent (ThermoFisher, Waltham, MA) for 24 hr. Cells were then washed with PBS and incubated with or without 10 µM sphingosine for 6 hr. Subsequently the NE-PER Nuclear Cytoplasmic Extraction Reagent kit (ThermoFisher, Waltham, MA) was used to generate nuclear and cytoplasmic fractions. Nuclear membrane sub-fractionation was accomplished using the Minute Nuclear Envelope Protein Extraction Kit (inVent). For western blotting, cellular samples were lysed in buffer (20 mM Tris pH 7.5,150 mM NaCl, 1% NP-40, with cOmplete Mini Protease Inhibitor Cocktail (Sigma) added immediately prior to use) and heated in 4x NuPage LDS Sample buffer and 10x NuPage Reducing Agent (both ThermoFisher, Waltham, MA) for 10 min at 95C. Proteins were resolved by electrophoresis on pre-cast 10% NuPage Bis-Trisgels and transferred to PVDF membranes using the iBlot system (both from Invitrogen). Membranes were blocked in 3% BSA in TBS with 1% Tween-20. Membranes were probed overnight with anti-Cers2 antibody (Sigma) at a 1:250 dilution, or with anti-GAPDH antibody (Abcam 9484) at 1:10,000 as a control for cytoplasmic proteins, anti-TATA Binding Protein (Abcam 51841) at 1:1000 as a control for nuclear proteins, and anti-LaminB1 (Abcam 16048) at 1:5000 as a control for nuclear envelope proteins. Blots were then probed with HRP-tagged secondary antibodies (Bio-Rad) and West Pico Chemiluminescence Reagent (Pierce) at 1:2000 in 5% milk-TBST for 1 hr at RT. Images were obtained and analyzed by c-DiGit Blot Scanner (LI-COR) and the corresponding Image Studio software.

## In situ hybridization

Whole mount in situ hybridization was performed as described (*Alexander et al., 1998*). Ovaries were removed from mutant or wildtype female adult fish, and oocytes gently dissociated in PBS. Following fixation in 4% paraformaldehyde (Sigma), hybridization was performed at 70C in 50% formamide buffer (Roche) with a digoxigenin-labeled RNA probe for *cers2b*, synthesized in vitro as an anti-sense strand using T7 polymerase from the BamH1-linearized cDNA clone.

## Acknowledgements

We thank Arielle Martel for excellent assistance with zebrafish husbandry, Ingrid Torregroza for in situ hybridization and Emily Mercer for confocal imaging. This study was supported by grants from the National Institutes of Health to TH (HL089934, CA077839, HL135821) and TE (HL111400). YH

was supported in part by postdoctoral fellowships from the Japan Society for the Promotion of Science Overseas Research Fellowships and by the Uehara Memorial Foundation.

## Additional information

### Funding

| Funder | Grant reference number | Author |
|---|---|---|
| Japan Society for the Promotion of Science | Overseas Research Fellowship | Yu Hisano |
| Uehara Memorial Foundation | | Yu Hisano |
| National Institutes of Health | HL089934 | Timothy Hla |
| National Institutes of Health | CA077839 | Timothy Hla |
| National Institutes of Health | HL135821 | Timothy Hla |
| National Institutes of Health | HL111400 | Todd Evans |

The funders had no role in study design, data collection and interpretation, or the decision to submit the work for publication.

### Author contributions

Karen Mendelson, Data curation, Formal analysis, Investigation, Writing—original draft, Writing—review and editing; Suveg Pandey, Yu Hisano, Data curation, Investigation, Writing—review and editing; Frank Carellini, Data curation, Formal analysis, Investigation; Bhaskar C Das, Resources, Data curation, Investigation; Timothy Hla, Conceptualization, Data curation, Formal analysis, Supervision, Funding acquisition, Writing—review and editing; Todd Evans, Conceptualization, Data curation, Formal analysis, Supervision, Funding acquisition, Methodology, Writing—original draft, Project administration, Writing—review and editing

### Author ORCIDs

Todd Evans (iD) https://orcid.org/0000-0002-7148-9849

### Ethics

Animal experimentation: This study was performed in strict accordance with the recommendations in the Guide for the Care and Use of Laboratory Animals of the National Institutes of Health. All of the animals were handled according to approved institutional animal care and use committee (IACUC) protocols (2011-100) of the Weill Cornell Medical College.

### Decision letter and Author response

Decision letter https://doi.org/10.7554/eLife.21992.011
Author response https://doi.org/10.7554/eLife.21992.012

## Additional files

### Supplementary files

• Transparent reporting form
DOI: https://doi.org/10.7554/eLife.21992.010

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
