## [Decision Letter]

Thank you for submitting your article "Cers2 Mediates Genomic Sensing and Regulation of Sphingosine Levels During Embryogenesis" for consideration by *eLife*. Your article has been favorably evaluated by Jonathan Cooper (Senior Editor) and three reviewers, one of whom is a member of our Board of Reviewing Editors. The reviewers have opted to remain anonymous.

The reviewers have discussed the reviews with one another and the Reviewing Editor has drafted this decision to help you prepare a revised submission.

Summary:

The manuscript reports a striking transcriptional up-regulation of maternal Cers2 in *sphk2^MZ^* embryos. Since the *sphk2^MZ^* embryos have elevated sphingosine levels, this suggested the possibility of a sphingosine-responsive element controlling the expression of Cers2, which could lower toxic sphingosine by catalyzing its conversion to ceramide. A sphingosine responsive element was subsequently defined in the Cers2 promoter. Intriguingly Cers2 itself was shown to was repress Cers2 promoter activity, leading to the suggestion the Cers2 functions as a transcription factor that feeds back to down regulate its own expression, an entirely novel mechanism for regulating sphingolipid metabolism.

Essential revisions:

1) Cers2 is a multi-pass membrane protein that resides in the ER. For it to function as a transcription factor, Cers2 or a fragment therein would have to move to the nucleus in order to bind to its promoter. Because of the novelty of Cers2 acting as a transcription factor and the alternative possibility described below, it is important to show that Cers2 enters the nucleus.

2) Another way that Cers2 could repress its own transcription is by lowering sphingosine levels through its intrinsic enzymatic activity, independent of an action as a transcription factor. Transfection of a catalytically dead mutant of Cers2 to see if it retains promoter repression activity could negate this possibility. Along a similar vein, the homeobox mutant (R121W, R122S) should be tested to see if it retains ceramide synthase activity.

3) Do zygotic *sphk2* mutants show higher levels of sphingosine and higher levels of *cers2b* transcripts (like the maternal-zygotic mutants)?

4) Are the effects of sphingosine on *cers2b* solely at the level of transcription? What about mRNA stability and/or translation?

5) Determine the levels of *cers2b* transcripts and ceramide in *sphk2* morphants as well as in wt embryos treated with sphingosine.

6) What are the comparative levels of Sph and ceramides in the *sphk2^MZ^* with and without the *cers2b* MO?

7) The extent of epiboly of EVL and Deep cells needs to be presented somehow, either as% progression or distance from the pole. Use of phalloidin staining could help demarcate these borders for measuring.

8) The authors conclude (quite reasonably) that the *sphk2^MZ^* mutants initiate a mechanism to reduce Sph levels during oogenesis, and this is protective. They also suggest that this is due to upregulation of *cers2b* during oogenesis. This needs to be demonstrated on the eggs squeezed from *sphk2* mutant females. Ceramide levels in oocytes should also be measured. Are maternal *sphk2* mutants also protected from exogenous Sph, as *sphk2^MZ^* mutants are?

---

## [Author Response]

Essential revisions:1) Cers2 is a multi-pass membrane protein that resides in the ER. For it to function as a transcription factor, Cers2 or a fragment therein would have to move to the nucleus in order to bind to its promoter. Because of the novelty of Cers2 acting as a transcription factor and the alternative possibility described below, it is important to show that Cers2 enters the nucleus.

This is a very good point and in response we carried out a number of experiments to investigate the cellular localization of Cers2, both at steady state and following exposure of cells to sphingosine. First, we used confocal microscopy of HEK293T cells transfected with a vector to express a Cers2-GFP fusion protein, which allowed us to visualize localization in live cells. As expected, this protein is localized primarily to the ER, throughout the cell including contiguous to the nuclear membrane. Following exposure to 10 μM sphingosine, a condition that leads to de-repression of the reporter that is regulated by the Cers2 promoter, there is indeed a considerable change in the distribution of the fusion protein and enhanced signal associated with the nuclear membrane. Whether this includes nuclear translocation of the fusion protein was not convincingly seen, but there did not appear to be a substantial incorporation into the nucleus. Second, we also compared expression levels of the protein by western blotting, following separation of nuclear and cystosolic extracts. In this case we found a marked enrichment for Cers2 in the nuclear fraction. However, when the extracts were further separated, we could show that within the nuclear fraction, the sphingosine induction caused Cers2 association primarily with nuclear membranes. Representative data from these experiments are presented in new Figure 6. In summary we did not find strong evidence that Cers2 functions as a nuclear protein and while it can’t be ruled out, we think it’s more likely to impact promoter activity by an indirect mechanism rather than as a nuclear transcription factor. Alternatively, modification of the Cers2b protein after sphingosine treatment (for example, proteolysis and release of an active fragment) cannot be determined as yet due to lack of appropriate antibodies. This will be addressed in future studies.

2) Another way that Cers2 could repress its own transcription is by lowering sphingosine levels through its intrinsic enzymatic activity, independent of an action as a transcription factor. Transfection of a catalytically dead mutant of Cers2 to see if it retains promoter repression activity could negate this possibility. Along a similar vein, the homeobox mutant (R121W, R122S) should be tested to see if it retains ceramide synthase activity.

As requested, we generated by site-directed mutagenesis, isoforms for Cers2b that are mutant in the Lag domain (HH212-213DD) that is predicted to be enzymatically dead, and compared the effect of this protein and the protein with a mutant HOX domain for Cers2 promoter regulation as well as regulation of ceramide levels. As shown in new panels for Figure 5, the Lag-mutant protein retains the ability to repress the reporter controlled by the Cers2b-promoter, yet fails to generate ceramide above background levels. In contrast, the HOX mutant protein retains the ability, equivalent to the wildtype protein, to increase ceramide levels, but is unable to repress activity of the promoter. Therefore, the HOX domain is required for Cers2 gene regulation, but independent of enzymatic function.

3) Do zygotic sphk2 mutants show higher levels of sphingosine and higher levels of cers2b transcripts (like the maternal-zygotic mutants)?

This is a challenging question to answer directly, because zygotic mutants are not identified by phenotype during early embryogenesis including the stages we evaluated in maternal-zygotic mutants. However, we could cross heterozygous females with either wildtype males or mutant males, to compare the relative levels of *cers2b* transcripts or sphingosine. In the former case, the embryos are 50% heterozygous and 50% wildtype (none are null), while in the latter case 50% of the embryos are zygotic null (and the other 50% heterozygous). If the zygotic null mutants were similar to MZ mutants, we would expect to easily measure half the levels of *cers2b* transcripts or increased sphingosine seen in the MZ mutants. However, we found the levels were not increased compared to the control cohort. Neither were overall S1P levels depleted. Therefore, the increased sphingosine levels associated with transcriptional activation of *cers2b* is dependent on the maternal-zygotic loss of *sphk2* (or in fact, as discussed below, maternal loss in oocytes).

4) Are the effects of sphingosine on cers2b solely at the level of transcription? What about mRNA stability and/or translation?

This is a possibility that we are not able to rule out, and have indicated so in the revised text (Discussion, fifth paragraph). However, we note that the *cers2b* transcript levels are markedly enhanced, at levels that are remarkably recapitulated by the promoter using a luciferase reporter with no shared RNA sequences. Furthermore, the increase is highly specific to the Cers2b transcript. These two observations make it unlikely that mRNA stability plays a major role.

5) Determine the levels of cers2b transcripts and ceramide in sphk2 morphants as well as in wt embryos treated with sphingosine.

As requested, we compared *cers2b* transcripts and ceramide levels in the *sphk2* morphants. As shown in new panels of Figure 4, *cers2b* fails to be up-regulated in the morphants, and neither do these embryos show increased ceramide levels. This suggests that the morphants are hypo-morphs for loss of S1P, and only in the mutant is the loss sufficient to cause sphingosine levels to rise and trigger the expression of *cers2b*. Alternatively, the activation of *cers2b* may be restricted to stages of oogenesis. Likewise, wildtype embryos treated with sphingosine fail to significantly increase levels of total ceramide and do not activate expression of *cers2b*. We also generated germline-transmitting mutants of *cers2b*, and found evidence that embryos maternal-zygotic for *sphk2* are now sensitive for loss of *cers2b*. In summary, the ability of embryos to activate *cers2b* expression is dependent on maternal-zygotic loss of *sphk2*, associated with enhanced levels of sphingosine, and *cers2b* is required under this condition to rescue gastrulation.

6) What are the comparative levels of Sph and ceramides in the sphk2^MZ^ with and without the cers2b MO?

As shown in Figure 1, *sphk2^MZ^* embryos are depleted of S1P, and have elevated levels of sphingosine and total ceramides. Knockdown of Cers2b (Figure 4) was sufficient to cause a gastrulation defect that phenocopies the defect associated with excess sphingosine (Figure 2). However, the morphant phenotype is not fully penetrant and we acknowledge that morphant knockdowns may be partial, transient, and subject to off-targets. Therefore, to more rigorously test whether Cers2b compensates for loss of Sphk2 to relieve excess sphingosine, we designed a sgRNA to target mutation of *cers2b*. We established a line of fish carrying a 23 bp deletion in exon6 of the *cers2b* gene that is predicted to delete most of the Lag domain and generate a frame-shift with a premature stop codon that would lack enzymatic function. This mutant allele is tolerated and was crossed onto the background of the *sphk2* mutant, to raise *sphk2^MZ^; cers2b+/-* adult fish. When crossed to each other, the resulting clutches consistently showed ~25% of the embryos phenocopied the gastrulation defect caused by treating wildtype embryos with excess sphingosine. These embryos very quickly perish, which precluded our ability to measure directly the ceramide levels. However, the genetic interaction of *sphk2* and *cers2b* is fully consistent with sphingosine-associated up-regulation and compensation from Cers2b.

7) The extent of epiboly of EVL and Deep cells needs to be presented somehow, either as% progression or distance from the pole. Use of phalloidin staining could help demarcate these borders for measuring.

As requested, we made measurements and indicate the percent of epiboly progression for representative embryos directly on the panels in Figure 3. We also note in the text (subsection “Epiboly defects caused by excess sphingosine are associated with a disturbed cytoskeleton”), the apparent loss of coordination for EVL and Deep cell attachment, as seen in panels J and K where the deep cells move independently of the EVL.

8) The authors conclude (quite reasonably) that the sphk2^MZ^ mutants initiate a mechanism to reduce Sph levels during oogenesis, and this is protective. They also suggest that this is due to upregulation of cers2b during oogenesis. This needs to be demonstrated on the eggs squeezed from sphk2 mutant females. Ceramide levels in oocytes should also be measured. Are maternal sphk2 mutants also protected from exogenous Sph, as sphk2^MZ^ mutants are?

As requested, we carried out in situ hybridization experiments using oocytes from *sphk2* mutant females. These experiments show strikingly higher levels of *cers2b* transcripts in the mutant oocytes. Similar to the *sphk2^MZ^* embryos, oocytes isolated from *sphK2* null female fish do not have measurable S1P, and display a commensurate accumulation of sphingosine. In this case, total ceramide levels are not significantly different from wildtype oocytes. Early embryos derived from a *sphk2* null female are resistant to rupture when exposed to exogenous sphingosine, despite a functional *sphk2* allele inherited from the wildtype male. All of these data are presented in new Figure 7, and are fully consistent with a sphingosine-induced Cers2b-based protective mechanism generated during oogenesis.